# Citrinin Provoke DNA Damage and Cell-Cycle Arrest Related to Chk2 and FANCD2 Checkpoint Proteins in Hepatocellular and Adenocarcinoma Cell Lines

**DOI:** 10.3390/toxins16070321

**Published:** 2024-07-17

**Authors:** Darija Stupin Polančec, Sonja Homar, Daniela Jakšić, Nevenka Kopjar, Maja Šegvić Klarić, Sanja Dabelić

**Affiliations:** 1OnkoLab Ltd., 10000 Zagreb, Croatia; info@onkolab.hr; 2University of Zagreb Faculty of Pharmacy and Biochemistry, Department of Biochemistry and Molecular Biology, 10000 Zagreb, Croatia; shomar@student.pharma.hr; 3University of Zagreb Faculty of Pharmacy and Biochemistry, Department of Microbiology, 10000 Zagreb, Croatia; daniela.jaksic@pharma.unizg.hr; 4Mutagenesis Unit, Institute for Medical Research and Occupational Health, 10000 Zagreb, Croatia; nkopjar@imi.hr

**Keywords:** citrinin, cell-cycle arrest, Chk2, FANCD2, lung adenocarcinoma, hepatocellular carcinoma, cytotoxicity, genotoxicity

## Abstract

Citrinin (CIT), a polyketide mycotoxin produced by *Penicillium*, *Aspergillus*, and *Monascus* species, is a contaminant that has been found in various food commodities and was also detected in house dust. Several studies showed that CIT can impair the kidney, liver, heart, immune, and reproductive systems in animals by mechanisms so far not completely elucidated. In this study, we investigated the CIT mode of action on two human tumor cell lines, HepG2 (hepatocellular carcinoma) and A549 (lung adenocarcinoma). Cytotoxic concentrations were determined using an MTT proliferation assay. The genotoxic effect of sub-IC_50_ concentrations was investigated using the alkaline comet assay and the impact on the cell cycle using flow cytometry. Additionally, the CIT effect on the total amount and phosphorylation of two cell-cycle-checkpoint proteins, the serine/threonine kinase Chk2 and Fanconi anemia (FA) group D2 (FANCD2), was determined by the cell-based ELISA. The data were analyzed using GraphPad Prism statistical software. The CIT IC_50_ for HepG2 was 107.3 µM, and for A549, it was >250 µM. The results showed that sensitivity to CIT is cell-type dependent and that CIT in sub-IC_50_ and near IC_50_ induces significant DNA damage and cell-cycle arrest in the G2/M phase, which is related to the increase in total and phosphorylated Chk2 and FANCD2 checkpoint proteins in HepG2 and A549 cells.

## 1. Introduction

Citrinin (CIT) is a polyketide mycotoxin with nephrotoxic properties produced by the *Penicillium*, *Aspergillus*, and *Monascus* species [1]. This contaminant has been found in various food commodities, including maize wheat, rye, barley, oats, rice, cereal-based products, fruits and fruit juices, roasted nuts, oilseeds, spices, cheese, fermented sausages, and feedstuffs [2]. Also, CIT has been detected in rice fermented with the red yeast *Monascus* spp., which is used for food coloring and meat preservation, as well as a dietary supplement [3]. Although CIT has been found in a variety of foodstuffs, no regulation limits have been set except for a safe limit for red rice-based food supplements. The limit of 100 µg/kg in rice fermented with red yeast *M. purpureus* has been set by European Union regulations [4,5]. Apart from CIT’s occurrence in foodstuffs, this mycotoxin was also detected in house dust [6,7], suggesting an uptake via the respiratory route as well.

Several studies showed that CIT can impair the kidney, liver, heart, immune, and reproductive systems in animals (reviewed in [1,8]). It has been classified as a group 3 compound by the International Agency for Research on Cancer [9]. According to an evaluation by the European Food Safety Authority [8], CIT induces renal adenomas in rodents but no carcinomas. Low concentrations of CIT reduce mitochondrial membrane potential, induce ROS-mediated DNA damage, and provoke calcium flux and cytochrome c release from mitochondria, which leads to cell apoptosis in liver and kidney cells [7,10].

The mechanism of CIT genotoxicity has not been elucidated completely. Studies on lung fibroblast hamster V79 cells and human liver HepG2 cells showed that CIT provoked aneuploidy and micronuclei formation without inducing DNA single-strand breaks, as concluded based on the tail length obtained by the comet assay [8,11]. In human peripheral blood mononuclear cells, CIT evoked chromosomal aberrations and activated G2/M cell-cycle arrest; these events were related to the inhibition of tubulin polymerization [12]. Recently, Wu et al. [13] showed that CIT induces G2/M cell-cycle arrest and cell injury in human L02 hepatocytes through ROS, induction of endoplasmic reticulum (ER), stress, and activation ofAMP-activated protein kinase (AMPK). 

The serine/threonine kinase Chk2 and Fanconi anemia (FA) group D2 (FANCD2) are two checkpoint proteins activated by single and/or double DNA strand breaks, leading to activation of effector proteins involved in cell-cycle regulation, p53 signaling, DNA repair, and apoptosis [14,15,16,17]. Due to DNA damage, both checkpoint proteins function as transducers of the serine/threonine protein kinase ATM (ataxia telangiectasia mutated) signal and become phosphorylated at Thr68 (ChK2) and Ser222 (FANCD2), which could result in cell-cycle arrest. We hypothesized that CIT applied in concentrations below IC_50_ in HepG2 and A549 cells might provoke DNA damage, alter Chk2 and FANCD2 phosphorylation and/or their expression, and induce cell-cycle arrest.

## 2. Results

### 2.1. Cytotoxicity

The results of the CIT effect on HepG2 and A549, obtained by MTT assay, are presented in Table 1.

The highest applied concentration of the vehicle control (i.e., DMSO) did not cause any significant changes in the cell metabolic activity, and the results of the CIT effect are expressed as a percentage of the vehicle control. Concentrations of CIT up to 50 µM did not cause significant changes, either in the HepG2 or A549 cell lines.

A concentration of 150 µM decreased the metabolic activity below 50% in HepG2 and IC_50_ for the HepG2 cells and was calculated to be 107.3 µM, while IC_30_ and IC_20_ were 96.4 and 76.0 μM, respectively (Figure 1a). The highest applied concentration of CIT did not cause a decrease in the metabolic activity by more than 50%, meaning that IC_50_ on A549 was >250 µM, while IC_30_ and IC_20_ were estimated to be 220.0 and 197.0 μM, respectively (Figure 1b).

### 2.2. Genotoxicity

The alkaline comet assay was used for the investigation of the genotoxic effect of CIT applied in concentrations below IC_50_ (determined by MTT assay) during 24 h on HepG2 and A549 cells (Figure 2).

The tail length (TL) was significantly increased for all CIT concentrations (i.e., 10, 50, and 100 µM) applied to the HepG2 cells compared to the control (vehicle-treated cells). In the A549 cells, only the highest concentration of CIT (200 µM) provoked a significant increase in the TL (Figure 2a).

Tail intensity (TI) was significantly higher in all CIT-exposed groups of both cell lines as compared to the control. In the HepG2 cells, an increase in the TI was dose-dependent. On the other hand, both 50 and 100 µM CITs caused quite the same statistically significant increase in TI on A549 cells, while the effect of 200 µM was several times stronger (Figure 2b).

### 2.3. Cell-Cycle Arrest

The cell-cycle distribution of HepG2 and A549 cells upon 24 h exposure to CIT applied in concentrations below IC_50_ was analyzed by flow cytometry. In HepG2 cells, 10 and 50 µM CIT did not significantly change cell-cycle distribution, while 100 µM CIT caused an accumulation of cells in the G2/M phase compared to the control (Figure 3a).

Consequently, decreases in cells in the S and G1 phases were also observed. Regarding A549, the smallest applied concentration of CIT (50 µM) showed no effect on the cell-cycle distribution, while an accumulation of cells in the G2/M phase and reduction of cells in the S phase was observed in both higher concentrations of CIT (100 and 200 µM). Additionally, exposure to 200 µM CIT also caused a decrease in the G1 phase (Figure 3b). As a control for cell-cycle perturbation, exposure to 275 nm UV for 5 min was applied.

### 2.4. Relative Chk2 and FANCD2 Phosphorylation and Protein Expression Level

The effects of the sub-IC_50_ concentrations of CIT on the level of total as well as phosphorylated forms of Chk2 and FANCD2 proteins were determined using a commercially available cell-based ELISA kit. The kit contained reagents for the determination of potential referent protein, GAPDH. However, our results showed that 200 µM CIT induced a significant increase (approximately 250% of the initial value) in GAPDH expression in A549 cells. Therefore, all the data were normalized, not to the level of GAPDH but to the number of viable cells detected by Crystal Violet staining.

For the HepG2 cells, neither 10 nor 50 µM CIT caused any significant changes in the total or phosphorylated forms of the tested proteins. However, the highest CIT concentration (100 µM) provoked significant and similar increases in both the total and phosphorylated forms of Chk2. Additionally, a comparable effect was observed regarding 100 µM CIT and FANCD2 protein—increases in both the total and phosphorylated forms of FANCD2 were detected (Figure 4a).

For the A549 cells, the smallest applied concentration of CIT (50 µM) caused no changes in the levels of total or phosphorylated forms of the tested proteins. Increases in both the total and phosphorylated forms of Chk2 protein were observed upon exposure to 100 µM CIT, and the intensity of changes was even stronger with the highest applied concentration of CIT, 200 µM. Regarding the FANCD2 protein, 100 µM CIT provoked small but statistically significant changes only in the phosphorylated form of FANCD2, while changes in both forms (total as well as phosphorylated) were detected as a result of exposure to 200 µM CIT (Figure 4b). Moreover, the intensity of change was more pronounced for the overall expression level of FANCD2 under 200 µM CIT exposure than for the phosphorylation of the same protein (Figure 4b).

## 3. Discussion

If consumed in sufficient quantities or by immunocompromised individuals, mycotoxins are poisonous to humans and other animals. To effectively combat the adverse impact posed by mycotoxins, primarily on health but also on the costs of the food and agriculture industries, extensive research is needed to determine the mechanism of action of mycotoxins. Citrinin, a secondary metabolite produced by fungi that contaminates long-stored food and can cause damage to various cells and organs, was in the spotlight of this research. Before exposure to CIT, the cells were serum-starved. Serum starvation prior to exposure to toxins/chemicals/drugs, etc., is a common procedure that allows the synchronization of cells to the same cell-cycle phase. In this regard, theoretically, the impact that the cell cycle will have on cells’ responses to treatment is removed. Therefore, it is easier to detect a toxin mode of action.

Several studies have shown that CIT is cytotoxic in a concentration-dependent manner to various cell lines, including Chinese hamster V79 lung fibroblasts (IC_50_ = 70 µM) [18], porcine kidney epithelial PK15 cells (IC_50_ = 73.5 µM) [19] human hepatocellular cancer Hep3B cells (IC_50_ = 124 µM) [20], human hepatocytes LO2 [13], and several others (reviewed in [21]). Cytotoxicity in mycotoxin research is measured by various methods (CKK-8 assay, AlamarBlue™assay, LDH cytotoxicity assay, and neutral red assay), but the most frequently used is the MTT assay, which was also used in this study. Our results also confirmed that CIT reduces cell viability in a concentration-dependent manner in both the HepG2 and A549 cell lines, but the HepG2 cells seem to be more sensitive to the CIT action compared to the A549 cells. Cytotoxicity of CIT in the HepG2 cells obtained in the present study (IC_50_ = 107.3 µM) is comparable with the previously reported IC_50_ (155 µM) for the same cell line [22]. However, the small discrepancies in the calculated IC_50_ can be explained by the different experimental procedures—namely, Gayathri et al. [22] calculated the IC_50_ for unsynchronized cells, while in the present study, cells were synchronized by serum starvation for 12 h prior to exposure to CIT. In the Alamar Blue assay performed by Johannessen et al. [23] on A549 cells, CIT provoked a reduction in cell viability to 40% upon 24 h of exposure to 200 µM. In our study, the maximum concentration of CIT (250 µM) leads to a decrease in viability to 53.6%. However, previous experiments of our research group showed that CIT causes cytotoxicity in A549 cells at twice as low a concentration (IC_50_ = 128 µM) if the cells are treated with the toxin without fetal bovine serum supplementation of RPMI medium [24]. This can be explained by the fact that it has been known for some time that CIT can bind to serum proteins [25,26,27], which may affect its availability for cell penetration and, consequently, result in a decrease in cytotoxicity.

Among several approaches for testing genotoxicity, the alkaline comet test was used in this study, and the results showed that CIT is genotoxic to both the HepG2 and A549 cells in concentrations below IC_50_. This genotoxicity assay measures both single and double DNA strand breaks [28], and both types of DNA damage could occur upon exposure to CIT. Taking into consideration the results of TI, which is the most useful parameter of the comet assay [28,29], we can conclude that all applied sub-cytotoxic concentrations provoked significant DNA damage in both cell lines; though, the HepG2 cells seem to be more sensitive to the genotoxic action of CIT. In human peripheral blood lymphocytes and V79 cells, CIT evoked micronuclei (MN) at concentrations of 10–100 µM and >30 µM, respectively [18,30]. In the HepG2 cells treated with 10–40 µM, CIT did not provoke a significant increase in the TL, but CIT-induced MN was predominantly centromere-positive, suggesting an aneugenic mode of action [11]. In the present study, in HepG2 cells, both the TL and TI were markedly increased using a CIT concentration below and approximately to IC_30_, which is in line with the findings of Gayathri et al. [22], who also showed that CIT provokes significant DNA damage in HepG2 cells in concentrations bellow IC_50_ (31 µM), suggesting a clastogenic mode of action. In the same study, a DNA fragmentation assay revealed that CIT induces double-strand DNA breaks, resulting in a DNA ladder [22]. Moreover, to the best of our knowledge, our findings are the first that showed a genotoxic effect on lung cancer A549 cells. Mitochondrial dysfunction by inhibition of the mitochondrial complex I, followed by an increased superoxide or hydroxyl radical accumulation, lipid peroxidation, modified antioxidative defense (e.g., decrease in glutathione levels), and inflammatory response in diverse eukaryotic cell models, have been linked to the genotoxicity of CIT (reviewed in [31]). Nevertheless, a subchronic exposure of male Wistar rats to a low dose of CIT (2 mg/kg b.w.) for 21 days showed that DNA damage in the liver and kidneys was partially related to oxidative stress, as obtained by the alkaline and hOGG1-modified comet assay [32]. Recently, gene-set enrichment analysis was carried out using GeneMANIA (gene interaction networks and pathway enrichment analyses) to discover the biological pathways and protein–protein interaction enrichments involved in CIT’s toxicity properties. The identified biological pathways possibly involved in genotoxicity were the signal transduction related to the DNA damage checkpoint, cellular responses to oxidative stress, chemical responses to oxidative stress, DNA damage responses to signal transduction by p53, and the stress-activated protein kinase signaling cascade [33]. According to the results of our study, CIT (100 µM) produces a significant accumulation of cells in the G2/M phase in both the HepG2 and A549 cells, which agrees with CIT-induced G2/M cell-cycle arrest in V79 (100 µM) and in human L02 hepatocytes treated with CIT at 50–450 µM [13,18]. CIT at a 50–100 µM concentration in human embryonic kidney (HEK293) cells has been found to activate the G2/M phase arrest with the inhibition of tubulin polymerization and mitotic spindle formation as well as chromosome aberration [12]. Additionally, cell-cycle arrest at the G0/G1 and G2/M phases, along with the induction of apoptosis, was also observed in CIT-treated mouse skin cells, which was accompanied by a significant increase in the expressions of p53, p21/waf1, Bax/Bcl-2 ratio, and cytochrome c release, as well as in the increased activities of caspase 9 and 3 [34]. Cell-cycle arrest enables the cells to check the level of DNA damage that might cause functional problems or lead to carcinogenesis; if the level of DNA damage leads to cell dysfunction, cell-cycle arrest proceeds apoptosis and cell death. DNA double-strand breaks provoke cell-cycle arrest either in G1/S or G2/M to avoid entering the S and M phases with damaged DNA [35].

Among the numerous checkpoint proteins involved in the regulation of cell-cycle progression, our attention in this study was focused on Chk2 and FANCD2. The expression profiles and phosphorylation statuses of these two proteins were measured by the cell-based ELISA. Based on the results of that method, the frequently used reference protein GAPDH was shown to be an inadequate choice for the normalization of CIT-exposure data since an overexpression of GAPDH was detected at high CIT concentrations. This finding should not be neglected, given that numerous researchers use GAPDH as a reference protein, assuming its stable expression, which consequently can result in the misleading interpretation of experimental data. Therefore, we normalized our experimental data to the number of viable cells detected by Crystal Violet staining, according to the option available within the kit. It is known that these two checkpoint proteins, both part of the ATM signaling pathway, are activated by phosphorylation, mainly at the positions Thr68 (Chk2) and Ser222 (FANCD2). Statistically significant changes in phosphorylation were observed at higher applied concentrations of CIT for both proteins as well as for both cell types. However, the observed changes, although statistically significant, are modest in intensity. However, a greater intensity of the changes related to phosphorylation is most often observed within a slightly shorter timeframe of exposure, while in this study, the effect of a one-day treatment was investigated. However, this 24 h exposure had an effect not only on phosphorylation but also on the total amount (both phosphorylated and non-phosphorylated forms) of proteins, which indicates that CIT influences some of the processes that affect the final amount of protein, either on a transcriptional, translational or perhaps a degradation level.

The two types of cancer cell lines used in this study are shown to respond differently to CIT exposure, and additionally, differences in the intensity of changes in (phospho)Chk2 and (phospho)FANCD2 were also observed. The hepatocellular carcinoma cell line, HepG2, had a lower IC50 compared to the adenocarcinoma cell line, A549, while the changes in phosphorylation and quantity of both Chk2 and FANCDC2 were more pronounced in the A549 cells. We can conclude that HepG2 is more sensitive to the action of CIT in such a way that its final effect is more harmful and deadly, while it is possible that the greater intensity of changes in these checkpoint proteins (observed in A549) makes these cells more resistant to the destructive effect of CIT, i.e., they show a greater capacity of DNA damage repair.

The review on the occurrence of CIT in various food commodities [36] showed a wide dissemination of CIT in different continents, such as Europe, Asia, America, and Africa. Cereal-based food and red rice food supplements seem to be the main source of exposure to CIT; the highest levels of CIT were found in maize in Serbia (mean 950 µg/kg), in Tom bran from Nigeria (up to 1173 µg/kg) and in red rice from China (up to 44,240 µg/kg). Apples that were analyzed in Portugal were also highly contaminated with CIT (320–920 µg/kg). In Germany, France, and Croatia, CIT was detected in cereals with mean concentrations of 10–30 µg/kg. However, CIT levels in food are insufficient for an intake assessment. Ali and Degen [37] compiled data on human exposure to CIT in different countries by conducting LC-MS/MS analyses of urine and plasma; in Belgium and Germany, concentrations in urine were much lower (0.016 and 0.03 ng/mL) than in Bangladesh (0.59 ng/mL) and Nigeria (5.96 ng/mL). The calculated provisional daily intake (PDI) of CIT in EU countries was lower than the tolerable daily intake (TDI, 0.2 μg/kg b.w.) set by EFSA [8], while in the Bangladesh and Nigerian cohorts, the PDI exceeded the TDI set for this mycotoxin. Data on the CIT levels in plasma were available for a very small number of subjects; the CIT concentrations were relatively low (0.06 ng/mL) in young adults from the Czech Republic and higher in subjects from Bangladesh (0.47 ng/mL). Based on the effective CIT concentrations applied in the in vitro models, it is difficult to predict the intracellular accumulation in target organs resulting from exposure to CIT in humans. The in vitro experiments using MDCK epithelial kidney cells indicated that CIT forms a stable complex with human serum albumin, which may influence the accumulation in target organs [27]. Yet, a half-life of CIT, estimated from the plasma concentration-time profiles in humans, indicated a short half-life of CIT (approximately 9 h) without a cumulative potential [38]. Taken together, humans are chronically exposed to CIT through food consumption, and occasionally, its levels may exceed the TDI, indicating that humans might be exposed to higher levels of CIT. Still, considering the available data on CIT in the plasma and urine of adults, it is unlikely that micromolar concentrations will be reached in target organs. Additionally, regarding oral intake, humans may be exposed to this mycotoxin through the respiratory route, as CIT was detected in house dust [6]. Some earlier studies have shown that inhalation of T-2 mycotoxin may be at least 10 times more toxic than an oral intake. [39,40]. The toxicological relevance of CIT intake through the respiratory route remains to be determined in future studies.

As in any study, one should be aware of the limitations of this type of research. One of the major drawbacks of the cytotoxicity assessment in cell lines is the lack of information on the interaction among different cell types, as well as on the impact of the immunological response on the toxin presence in complex organisms. Also, more informative insights could be obtained by conducting experiments in a wider timeframe. Confirmation of specific proteins that are involved in the mechanism of action of a particular toxin could be obtained by using inhibitors of particular protein/signaling pathways with simultaneous treatment with mycotoxins.

## 4. Conclusions

In conclusion, CIT in concentrations below and near IC_50_ induces significant DNA damage and cell-cycle arrest in the G2/M phase, which is related to an increase in the total and phosphorylated Chk2 and FANCD2 checkpoint proteins in HepG2 and A549 cells. Since both checkpoint proteins are important participants in the repair of DNA double-strand breaks [16,17], our study indicates that CIT action involves, but is probably not limited to, the induction of DNA double-strand breaks. Decreased cell survival at higher concentrations of CIT shows that despite the increase in the amount of Chk2 and FANCD2, a significant number of cells fail to repair such breaks and proceed to cell death, and the capacity to overcome the toxic effects is also cell-type-dependent.

## 5. Materials and Methods

### 5.1. Reagents

Components for the cell culture maintenance, including RPMI 1640, FBS, trypsin-EDTA, phosphate-buffered saline (PBS; Ca^2+^ and Mg^2+^ free), penicillin, and streptomycin, were from Lonza (Basel, Switzerland). CIT was purchased from Cayman Chemicals, Ann Arbor, MI, USA), while NaOH and NaCl, used for the comet test, were from Kemika (Zagreb, Croatia). The colorimetric cell-based ELISA kits CytoGlow™ Chk2 (phosphoThr-68) and CytoGlow™ FANCD2 (phosphoSer-222) were from AssayBioTech (Fremont, CA, USA). If not stated otherwise, all the other reagents were purchased from Sigma-Aldrich (Steinheim, Germany).

### 5.2. Cell Cultures

HepG2 (human hepatocellular carcinoma) and A549 (human lung adenocarcinoma) were obtained from the European Collection of Cell Cultures (Salisbury, UK). The cells were grown in complete cell culture media consisting of RPMI 1640, 2 mmol/L glutamine, 10% heat-inactivated fetal bovine serum (FBS), penicillin (100 IU/mL; 1 IU 67.7 µg/mL), and streptomycin (100 µg/mL). The HepG2 and A549 cells were grown in a moisturized atmosphere with 5% CO_2_ at 37 °C and 95% relative humidity. All experiments were performed on the cells between passages 4 and 7.

### 5.3. MTT Proliferation Assay

The CIT stock solution was prepared in cell-culture-grade dimethyl sulfoxide (DMSO) in a concentration of 0.03 M. The HepG2 and A549 cells were seeded in a 96-well flat-bottom microplate in a complete medium (10^4^ cells/well). Following 24 h of incubation in a complete medium, the cells were serum-starved for 12 h. Afterward, the medium was aspirated, and complete media containing different concentrations of CIT were added to each well. The CIT concentrations ranged from 0.1 to 200 µM for HepG2 cells or from 0.1 to 250 µM for A549 cells. The controls contained the test model cells and culture medium (containing either the complete medium or the DMSO in the highest final concentration, 0.33%) but no CIT. Upon 24 h exposure, the medium was removed, and in each well, 100 µL of MTT tetrazolium salt reagent [3-(4,5-dimethylthiazol-2-yl)-2,5-diphenyltetrazolium bromide] diluted in RPMI 1640 (0.5 mg/mL) was added. Following a 3 h incubation, the medium was discarded, and 150 µL of DMSO was added to each well to dissolve the formed formazan crystals. The cells were incubated at room temperature on a rotary shaker for 10 min. The absorbance was measured using a microplate reader SpectraMax i3x (Molecular Devices, San Jose, CA, USA) at a wavelength of 540 nm. A negative control (medium without cells) was used as a blank. The concentrations that inhibited the viability of 50% of the cells (IC_50_) were determined by nonlinear regression analysis, described in Statistical Analysis. All tests were performed in three biological replicates (each biological replicate in 5 technical replicates), and the results were expressed as a percentage of the control cells (C).

### 5.4. Alkaline Comet Assay

The methodology was performed according to Sing et al. [41], with some modifications. The cells were seeded in 6-well plates (2 × 10^5^ cells/well), incubated for 24 h in a complete medium, serum-starved for 12 h, and treated with sub-IC50 concentrations of CIT diluted in a complete medium for the subsequent 24 h. More precisely, the HepG2 cells were treated with 10, 50, and 100 µM CIT, while the A549 cells were treated with 50, 100, and 200 µM CIT. The controls contained the test model cells and culture medium (containing either the complete medium or the DMSO in the highest final concentration, 0.33%) but no CIT. Upon incubation and washing with cold PBS, the cells were trypsinized. The cell suspensions were prepared in fresh, complete media and centrifuged (200 g, 2 min). The cell pellets were resuspended in fresh media, and 50 µL of the cell suspension was mixed with 0.5% LMP agarose. The samples were spread on slides pre-coated with 1% and 0.6% NMP agarose and solidified on ice for 10 min. Both the LMP and NMP agaroses were prepared in Ca- and Mg-free PBS. The prepared slides were subjected to an alkali lysis solution (pH 10) containing NaCl (2.5 mol/L), Na_2_EDTA (100 mmol/L), Tris (10 mmol/L), and Triton-X 100 (1%). After the lysis (+4 °C, 60 min), the slides were immersed in denaturation alkaline buffer (NaOH 10 mmol/L, Na_2_EDTA200 mmol/L, pH 13) for 20 min, and the same type of buffer was used for electrophoresis (25 V and 300 mA, 20 min). The neutralization was performed by dripping the slides with Tris/HCl buffer (0.4 mol/L, pH 7.5). The slides were stained with ethidium bromide (20 µg/mL). All experiments were performed in two biological replicates (each in two technical replicates). In each biological replicate, images of 200 randomly selected nucleoids (100 nucleoids from each of the two technical replicates) were observed by fluorescence microscope (Olympus, Tokyo, Japan) and analyzed using the Comet assay IV software package (InstemPerceptive Instruments Ltd., Suffolk, Halstead, UK). Only comets with a defined head were scored. Tail length (presented in micrometers) and tail intensity (i.e., percentage of DNA in the comet tail) were selected as the measurable indicators of DNA damage, as described in Colins [28].

### 5.5. Flow Cytometry Analysis

The flow cytometry experiments and analyses were performed as a commercially available service in BIOCentre (Incubation Centre for Biosciences, Zagreb, Croatia). Briefly, the HepG2 and A549 cells were incubated in 6-well plates (105 cells/mL) in RPMI 1640 medium (with 10% of FBS) for 24 h. Following the 24 h incubation, the cells were synchronized as previously described and then treated for 24 h with CIT (10, 50, and 100 µM for HepG2, and 50, 100, and 200 µM for A549) dissolved in a complete medium. The control cells were exposed to the complete media or the highest concentration of vehicle DMSO. The 5 min exposure to UV light (275 nm) was applied as a control for cell-cycle perturbation. The preparation of samples for flow cytometry was completed as described in Klasić et al. [42]. Upon exposure, the cells were trypsinized and washed in PBS. After washing, the cells were resuspended in 1 mL of PBS and fixed with 3 mL of cold absolute ethanol. Prior to flow cytometry analysis, the cells were washed with PBS and then treated with RNase A (20 μg/mL) for 30 min at 37 °C. After incubation, the cell samples were stained with propidium iodide (50 μg/mL) and analyzed on a BD FACSVerseTM Flow Cytometer (San Jose, CA, USA). FlowJo software version 10.8 (TreeStarInc. Ashland, OR, USA) was used to determine the percentages of cells in each phase of the cell cycle (G1, S, G2/M). The analyses were performed on two biological replicates (each in two technical replicates).

### 5.6. Chk2 and FANCD2 Determination by Colorimetric Cell-Based ELISA

The colorimetric cell-based ELISA kits CytoGlow™ Chk2 (phosphoThr-68) and CytoGlow™ FANCD2 (phosphoSer-222) were used to determine the relative phosphorylation and expression profiles of Chk2 and FANCD2 in the HepG2 and A549 cells. The analysis was performed according to the product manual. Briefly, 2 × 10^4^ cells/well were seeded in 96-well plates in complete media, left for 24 h to adhere, and then serum-starved for 12 h. The HepG2 and A549 cells were then treated for the next 24 h with CIT (10, 50, and 100 µM, and 50, 100, and 200 µM, respectively) and prepared in the complete media. The control cells were exposed to the complete media or the highest concentration of vehicle DMSO. Following incubation, procedures recommended by the manufacturer were performed, which included fixation in 4% formaldehyde, quenching of endogenous peroxidase activity, blocking, incubation with primary antibodies (against the phosphorylated form or whole protein, regardless of the phosphorylation status), incubation with horse-radish-peroxidase-conjugated secondary antibodies, which included peroxidase substrate addition, stopping the peroxidase reaction, and cell-washing between steps. The absorbance was measured at a wavelength of 450 nm using a microplate reader SpectraMax i3x (Molecular Devices). As a potential referent protein for which the reagents were provided in the kit, GAPDH was tested. For normalization purposes, also included in the kit, a Crystal Violet cell stain that provides absorbance readings proportional to the cell count, was applied as the next step. The procedure included washing of the cells, incubation with stain, washing of unbound stain, and solubilization of the stain with SDS-solution, followed by the absorbance reading at 590 nm. All experiments were performed in two biological replicates (each in four technical replicates). No significant changes were observed between the control cells incubated in complete media compared to the cells incubated with vehicle DMSO. Considering CIT had induced observed changes in the GAPDH expression level, the results were finally expressed as a percentage of control and normalized to the Crystal Violet stain-determined cell number.

### 5.7. Statistical Analysis

The GraphPad Prism software version 9.0.0 for Windows (GraphPad Software, SanDiego, CA, USA) was used for the determination of the inhibitory concentration as well as for statistical analyses. The normality of the data distribution was accessed using the Kolmogorov–Smirnov and Shapiro–Wilk tests. The IC_50_ values (concentrations that reduced cell growth to 50% of the initial value) were determined using a nonlinear regression curve fit. Analysis of variance (ANOVA) followed by the post hoc Bonferroni test was used to assess the significant differences among groups. Prior to the analysis of the comet assay results, the tail intensity (TI) and tail length (TL) were log10 transformed to achieve normal distribution. Additionally, due to the presence of zero values within the TI, prior to log10 transformation, all TI values were increased by 1. The results are shown as the mean values ± standard error of the mean (SEM). All *p*-values less than 0.05 were considered statistically significant.

## Figures and Tables

**Figure 1 toxins-16-00321-f001:**
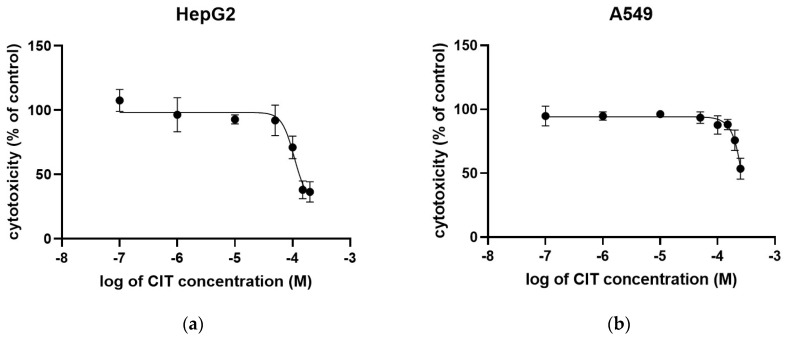
Determination of IC_50_ values of CIT on (**a**) HepG2 and (**b**) A549 cells. Nonlinear regression analysis of the average metabolic activity values measured by MTT assay of HepG2 and A549 cells after 24 h exposure to various concentrations of CIT. Each data point represents the mean ± SEM of three independent experiments performed in hexaplicates.

**Figure 2 toxins-16-00321-f002:**
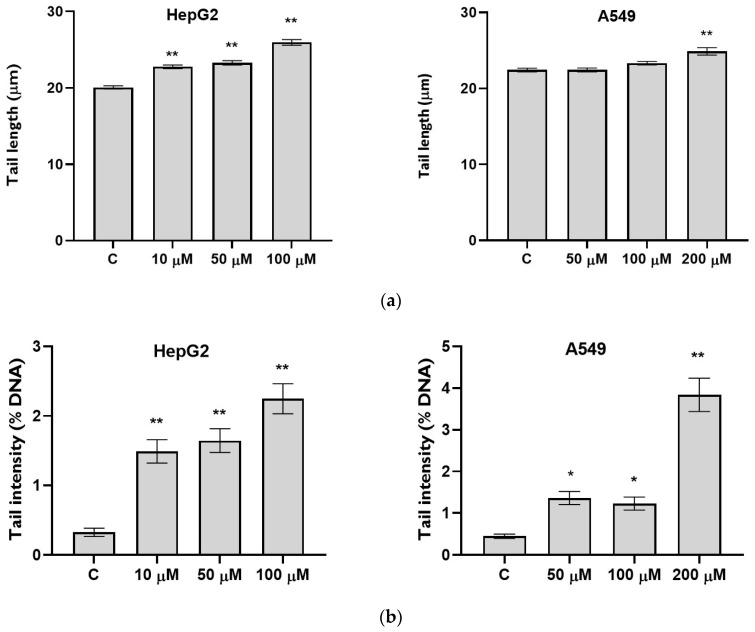
Genotoxic effect of CIT on HepG2 and A549 cells, measured by alkaline comet assay and presented as (**a**) tail length and (**b**) tail intensity. Cells were exposed to sub-IC_50_ concentration of CIT for 24 h. The results are presented as a back-transformed mean of log10 TL and TI ± SEM. Control cells were cultivated in complete media with the highest applied concentration of vehicle DMSO. * *p* < 0.01, ** *p* < 0.001.

**Figure 3 toxins-16-00321-f003:**
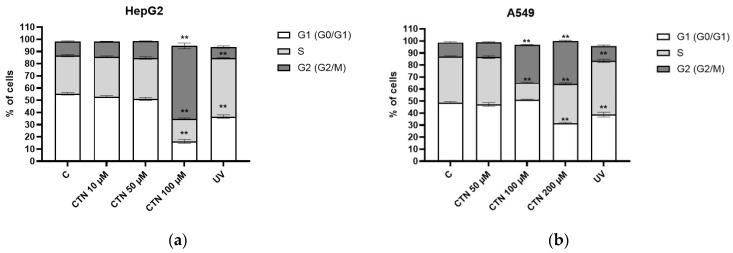
The effect of CIT on the cell-cycle distributions of (**a**) HepG2 and (**b**) A549 cells, determined by flow cytometry. Cells were exposed to sub-IC_50_ concentration of CIT for 24 h. The results are presented as a percentage of cells in a particular cell-cycle phase (G0/G1, S, and G2/M) as mean ± SEM. Control cells were cultivated in complete media with the highest applied concentration of vehicle DMSO, while exposure to UV (275 nm for 5 min) was used as a flow cytometry control. ** *p* < 0.001.

**Figure 4 toxins-16-00321-f004:**
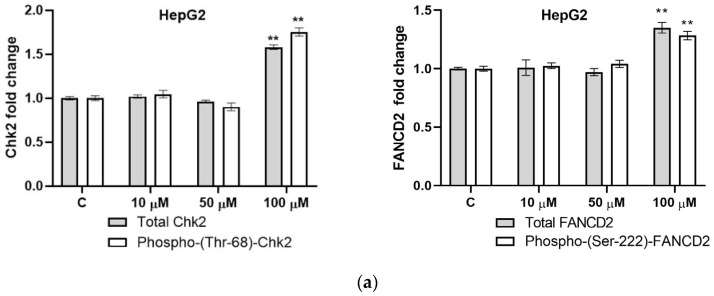
The effect of CIT on relative levels of total and phosphorylated Chk2 and FANCD2 checkpoint proteins in (**a**) HepG2 and (**b**) A549 cells, determined by CytoGlow colorimetric cell-based ELISA kits. Cells were exposed to sub-IC_50_ concentration of CIT for 24 h. The results of the immune-Cyto-Glow ELISA were normalized to the number of viable cells and expressed as a fold change compared to the control cells, mean ± SEM. Control cells were cultivated in complete media with the highest applied concentration of vehicle DMSO. * *p* < 0.01, ** *p* < 0.001.

**Table 1 toxins-16-00321-t001:** Cytotoxicity of HepG2 and A549 cells after 24 h exposure to CIT.

CIT (µM)	Cell Viability of HepG2 (% of Control)	Cell Viability of A549 (% of Control)
0.1	107.6 ± 3.8	95.0 ± 3.1
1	96.4 ± 5.9	95.0 ± 1.3
10	92.8 ± 1.6	96.3 ± 1.0
50	92.0 ± 5.3	93.5 ± 2.0
100	71.0 ± 4.0 **	88.0 ± 2.9 *
150	38.0 ± 3.0 **	88.1 ± 1.6 *
200	36.4 ± 3.5 **	75.8 ± 3.6 **
250	-	53.6 ± 3.6 **

Data are given as the mean ± standard error of the mean (SEM) from three independent experiments. Control cells were cultivated in complete media with the highest applied concentration of vehicle DMSO. * *p* < 0.01, ** *p* < 0.001.

## Data Availability

The original contributions presented in this study are included in the article. Additional information is available from the corresponding authors upon reasonable request.

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
