# Peer review of "Citrinin Provoke DNA Damage and Cell-Cycle Arrest Related to Chk2 and FANCD2 Checkpoint Proteins in Hepatocellular and Adenocarcinoma Cell Lines"

_toxins, 2024, doi:10.3390/toxins16070321_

Round 1

Reviewer 1 Report

Comments and Suggestions for Authors

The article describes the effects of the mycotoxin Citrinin (CIT) on two cell lines HepG2 (hepatocellular carcinoma) and A549 (lung adenocarcinoma).  However, in my opinion, the number of experiments conducted is not enough to draw conclusions and the manuscript is not ready to be published in toxins. Nevertheless, some of the comments are included below:

- name of the species should be written in italics (lines 6 and 7)

- please explain why you decided to use these particular cell lines because it is not clear to me

- the resolution of the figure 2 is quite poor

-I will suggest adding to the figure 2 a representative photo of comets 

Author Response

Citrinin provoke DNA damage and cell cycle arrest related to Chk2 and FANCD2 checkpoint proteins in hepatocellular and adenocarcinoma cell line

We hugely thank the reviewers and the editor for giving us the opportunity to respond to the reviewers’ suggestions, correct the manuscript accordingly and give the improved version for the reconsideration for publication in Toxins. We are grateful to all the reviewers for taking the time to review this manuscript. We have taken into consideration all the kind suggestions to improve the quality of the paper. Please find the detailed responses below and the corresponding revisions/corrections highlighted in the re-submitted files.

Point-by-point response to Comments and Suggestions for Authors - Reviewer 1:

- name of the species should be written in italics (lines 6 and 7)

We apologize for this formal mistake and have corrected that as suggested.

- please explain why you decided to use these particular cell lines because it is not clear to me

This study was performed on two human cell lines of different origin, human hepatocellular carcinoma HepG2 cells and human lung adenocarcinoma A549 cell line. This research is a continuation of our previous research on the effects of different toxins or combinations of toxins on human cells. We believe that studies performed on the cells of the same organism with different morphological and functional features can provide valuble data that can help reveal mechanisms of action of particular toxin.

- the resolution of the figure 2 is quite poor

Thank you for pointing this out as indeed the resolution of the Fig. 2 was quite poor. We inserted new version of the Figure with better resolution.

(a)

(b)

-I will suggest adding to the figure 2 a representative photo of comets 

The alkaline comet test is a screening test for DNA damage in different types of cells, from human lymphocytes as a primary culture, to cell lines of human or animal origin. The photo would not show any specific difference in the appearance of A549 and HepG2 comets for e.g. a tail intensity value of 1 and it is not a common practice to insert photos of comets in articles. Therefore, to our opinion, adding an image would not contribute to the quality of the presentation of the results, and we hope that it is not necessary to accept this suggestion.

Reviewer 2 Report

Comments and Suggestions for Authors

General remarks

Two tumour cell lines, i.e. HepG2 and A549, were treated with citrinin (CIT) for 24 h with a range of concentrations, e.g. 0.1 and 250 µM CIT to assess cytotoxicity by MTT assay.  Then 10 to 200 µM CIT were applied in the alkaline comet assay and to study cell cycle distribution by FACS analysis and expression of the total and phosphorylated cell cycle proteins Chk2 and FANCD2 by ELISA assays.

In short, standard assays were applied to study concentration dependent effects of CIT in two cell lines for the studied endpoints (cytotoxicity, DNA strand breaks, cell cycle arrest, and expression of two checkpoint proteins). Results are reported in a suitable way, yet this can be improved.

However, the author’s interpretation of their data is only partly adequate, and the study design also raises questions (see below specific comments). The manuscript has to be revised taking into account several points of critique and further corrections.

Specific comments

1.     Cytotoxicity for 24h CIT treatment and reported as IC50 value, was 107.3 µM in HepG2 cells and ≥ 250 µM in A459 cells.  But, it is desirable to provide also IC20 or IC30 values that will indicate a lower, but notable degree of cytotoxicity. Such values can be derived from Fig. 1.

2.     The CIT levels tested in comet assays, flow cytometry and protein analysis are denoted as “sub-IC50 concentrations“; yet treatment concentrations which led to notable changes, e.g. in cell cycle distribution were 100 and 200 µM CIT, i.e. close to IC50 values in HepG2 and A459 cells. Thus, legends of Figs. 2, 3 and 4 should better state: Cells were exposed up to the IC50 concentration of CIT for 24 hours.

3.     Regarding the comet assay, the referee is asking why this endpoint was chosen in the first place, since several studies on the genotoxicity of CIT revealed induction of micronucleated cells, predominantly by an aneugenic mechanism and interference with microtubuli (Refs. [10; 11; 18] and Pfeiffer et al. 1998 cited e.g. in [7 or 11] which occurred at concentrations without notable decreases in cell viability.

4.     Clastogenic effects (DNA strand breaks) as detected by comet assays in the present study occurred mainly at CIT concentrations which were also affecting cell viability; and CIT levels close to IC50 values had marked effects on cell cycle distribution of HepG2 and A459 cells.

5.     Accumulation of cells in G2/M phase is not necessarily related to DNA strand breaks alone; and the discussion of CIT genotoxicity should not be limited to its clastogenic potential.  

6.     The y-axis in Fig. 4 states “% of control“; but this is wrong since the expression of proteins related to control is usually given as x-fold change (not %).

7.     With numerous checkpoint proteins being involved in cell cycle progression, it is not really clear why the authors have focused on Chk2 and FANCD2?

8.     Differences in the response to CIT treatment between cell lines is not a new observation (as reviewed in [17 and 20]). Although the present study may be the first to report CIT effects on the lung cancer cell line A459, it is not the first study with lung (i.e. V79) cells.

9.     The „limitations of this type of research“ (line 262 and ff. text) is not really adequate (and mainly speculative). Instead, the Discussion should also address other published data on the genotoxic potential of CIT (see above remark #3).

10.  What is missing also is a reflection how concentrations tested in vitro relate to in vivo CIT levels reported in plasma or serum of mycotoxin-exposed humans. Such data can be found in biomonitoring studies as reviewed e.g. by Ali N et al. doi: 10.1007/s00204-019-02570-y) and the last Discussion section of an earlier in vitro study (Ref. [17]).

Corrections

Abstract, line 10: delete ’culture’ before lines

Line 18: (edit) ... CIT at levels near the IC50 values induces ...

Line 66 and heading of Table 1: It can be argued whether „Metabolic activity“ is the best term or rather „Cytotoxicity“ ...

Fig. 2 Legend should state that data for TL and TI were log10 transformed.

Lines 109, 127, 129 and 131: replace „faze“ by the correct term „phase“ .......

Line 156: ...CIT-treatment, than (not: then) .....

Line 207: correct the spelling of „below“ (not: bellow)

Line 272 (edit): ... concentrations near and sometimes below IC50 values

Comments on the Quality of English Language

by and large o.k. corrections stated in report

Author Response

Citrinin provoke DNA damage and cell cycle arrest related to Chk2 and FANCD2 checkpoint proteins in hepatocellular and adenocarcinoma cell line

We hugely thank the reviewers and the editor for giving us the opportunity to respond to the reviewers’ suggestions, correct the manuscript accordingly and give the improved version for the reconsideration for publication in Toxins. We are grateful to all the reviewers for taking the time to review this manuscript. We have taken into consideration all the kind suggestions to improve the quality of the paper. Please find the detailed responses below and the corresponding revisions/corrections highlighted in the re-submitted files.

Point-by-point response to Comments and Suggestions for Authors - Reviewer 2:

Specific comments

  1. Cytotoxicity for 24h CIT treatment and reported as IC50 value, was 107.3 µM in HepG2 cells and ≥ 250 µM in A459 cells.  But, it is desirable to provide also IC20 or IC30 values that will indicate a lower, but notable degree of cytotoxicity. Such values can be derived from Fig. 1.

Thank you for this suggestion. We calculated IC20 and IC30 and added that information into the section Results.

“Concentration of 150 µM decreased metabolic activity below 50% in HepG2 and IC50 for HepG2 cells was calculated to be 107.3 µM, while IC30 and IC20 were 96.4 and 76.0 μM, respectively (Figure 1a). The highest applied concentration of CIT did not cause decrease of metabolic activity more than 50%, meaning that IC50 on A549 was > 250 µM, while IC30 and IC20 were estimated to be 220.0 and 197.0 μM, respectively (Figure 1b).

  1. The CIT levels tested in comet assays, flow cytometry and protein analysis are denoted as “sub-IC50 concentrations“; yet treatment concentrations which led to notable changes, e.g. in cell cycle distribution were 100 and 200 µM CIT, i.e. close to IC50 values in HepG2 and A459 cells. Thus, legends of Figs. 2, 3 and 4 should better state: Cells were exposed up to the IC50 concentration of CIT for 24 hours.

It is worth mentioning that two cell lines were not exposed to the same concentration of CIT, namely, the highest applied concentration of CIT for HepG2 cells was 100 µM, while A549 were exposed to 200 µM CIT. Taking into consideration that calculated IC50 values for HepG2 was 107.3 µM and that HepG2 cells were exposed to 10, 50 and 100 µM CIT, as well as the fact that IC50 for A549 that was >250 µM and A549 cells were exposed to 50, 100 and 200 µM CIT, we believe that those values can be called sub-IC50 values.

We combined the answers to reviewer 1's comments No 3, 4 and 5, because the issues mentioned in the comments are interconnected. In the text that follows, we hope we are giving an adequate explanation of the given questions, followed by the intended changes in the manuscript.

  1. Regarding the comet assay, the referee is asking why this endpoint was chosen in the first place, since several studies on the genotoxicity of CIT revealed induction of micronucleated cells, predominantly by an aneugenic mechanism and interference with microtubuli (Refs. [10; 11; 18] and Pfeiffer et al. 1998 cited e.g. in [7 or 11] which occurred at concentrations without notable decreases in cell viability.
  2. Clastogenic effects (DNA strand breaks) as detected by comet assays in the present study occurred mainly at CIT concentrations which were also affecting cell viability; and CIT levels close to IC50 values had marked effects on cell cycle distribution of HepG2 and A459 cells.
  3. Accumulation of cells in G2/M phase is not necessarily related to DNA strand breaks alone; and the discussion of CIT genotoxicity should not be limited to its clastogenic potential.  

This research was intended as assessment of the DNA damaging potency of CIT. That was of particular interest regarding A549 cells since to our knowledge there is no data on CIT genotoxic activity in A549 cells obtained by comet assay. The concentrations were chosen based on MTT test. These concentrations were below the IC50 and do not cause decrease in cell viability for more than 30% comparing to control, as recommended for performing the comet test (Tice et al. 2000. Single Cell Gel/Comet Assay: Guidelines for In Vitro and In Vivo Genetic Toxicology Testing. Environmental and Molecular Mutagenesis 35:206-221.). The highest concentration used in the comet (100 µM in HepG2, 200 µM in A549) reduced cell viability to 71 and 75%, respectively (Table 1). The calculated IC30 (HepG2 96.4 µM, A549 220 µM), suggested by the reviewer, are close to cell viability values obtained by MTT assay. The alkaline comet assay is considered one of the elementary tests for screening and early assessment of DNA damaging effects which notes reversible DNA damage (and irreversible at high DNA tail intensity). Usually, based on the results of such assays a decision can be made about wider research; we believe that in this study the use of the comet assay is justified as a screening that precedes the investigation of the mechanisms involved in DNA damage repair signalling. Although there are already data for the comet assay on HepG2 cells in the literature cited in the manuscript under 21 (Gayahtiri et al., 2015), we believe that the comet should have been made under the same experimental conditions as was done for A549 cells to compare the results of the comet and to further discuss on impact of CIT on checkpoint proteins. We agree with the reviewer regarding comment at point 9., we expanded the discussion on genotoxic potential of CIT as follows:

In human peripheral blood lymphocytes and V79 cells, CIT evoked micronuclei (MN) at concentrations 10-100 µM and > 30 µM, respectively [18,30]. In HepG2 cells treated with 10-40 µM, CIT did not provoke significant increase in TL but CIT-induced MN were predominantly centromere positive suggesting aneugenic mode of action [11]. In the present study in HepG2 cells, both TL and TI were markedly increased at CIT concentration below and approximately to IC30, which is in line with the findings of Gayathri et al [22], who also showed that CIT provokes significant DNA damage in HepG2 cells in concentrations bellow IC50 (31 µM), suggesting clastogenic mode of action. In the same study, DNA fragmentation assay revealed that CIT induces double strand DNA breaks resulting in DNA ladder [22]. Moreover, to the best of our knowledge, our findings are the first which showed genotoxic effect on lung cancer A549 cells. Mitochondrial dysfunction by inhibition of the mitochondrial complex I followed by increased superoxide or hydroxyl radical accumulation, lipid peroxidation, modified antioxidative defence (e.g. decrease in glutathione levels) and inflammatory response in diverse eukaryotic cell models have been linked to genotoxicity of CIT (reviewed in [31]). Nevertheless, subchronic exposure of male Wistar rats to low dose of CIT (2 mg/kg b.w.) for 21 days showed that DNA damage in liver and kidney was partially related to oxidative stress, as obtained by alkaline and hOGG1-modified comet assay [32]. Recently, gene set enrichment analysis was carried out using GeneMANIA (Gene Interaction Networks and Pathway Enrichment Analysis) to find biological pathways and protein–protein interaction enrichments involved in CIT toxicity properties. Identified biological pathways, possibly involved in genotoxicity, were signal transduction related to DNA damage checkpoint, cellular response to oxidative stress, chemical response to oxidative stress, DNA damage response signal transduction by p53 and stress-activated protein kinase signalling cascade [33]. According to the results of our study, CIT (100 µM) produces significant accumulation of cells in G2/M phase, in both HepG2 and A549 cells, which agrees with CIT-induced G2/M cell cycle arrest in V79 (100 µM) and in human L02 hepatocytes treated with CIT at 50-450 µM [13,18]. CIT at 50-100 µM concentration in human embryonic kidney (HEK293) cells has been found to activate G2/M phase arrest with inhibition of tubulin polymerization and mitotic spindle formation as well as chromosome aberration [12]. Additionally, cell cycle arrest at the G0/G1 phase and G2/M phase, along with the induction of apoptosis, was also observed in CIT-treated mouse skin cells, which was accompanied with a significant increase in the expression of p53, p21/waf1, Bax/Bcl-2 ratio and cytochrome c release, as well as in increased activities of caspase 9 and 3 [34]. Cell cycle arrest enables cells to check the level of DNA damage that might cause functional problems or lead to carcinogenesis; if the level of DNA damage leads to cell dysfunction, cell cycle arrest proceeds apoptosis and cell death. DNA double strand breaks provoke cell cycle arrest either in G1/S or G2/M to avoid entering S and M phase with damaged DNAs [35].

  1. The y-axis in Fig. 4 states “% of control“; but this is wrong since the expression of proteins related to control is usually given as x-fold change (not %).

Thank you for this observation on the obvious mistake regarding the name of the axis. Values presented on the Fig. 4 were correctly presented as a fold change, only the name of the axis was wrong. We changed it to “fold change” and inserted new version of the Fig. 4, as shown below.

(a)

(b)

With numerous checkpoint proteins being involved in cell cycle progression, it is not really clear why the authors have focused on Chk2 and FANCD2?

Cell cycle progression is indeed sophisticated and highly regulated process which, among many other molecules, includes various checkpoint proteins. Those proteins are part of the complex pathways, some of which are more, and some less studied. The serine/threonine kinase Chk2 is the transducer kinase involved in spreading the DNA damage signal through the phosphorylation of effector proteins involved in DNA repair, cell cycle regulation, p53 signalling, and apoptosis. Chk2 is phosphorylated at Thr68 by serine/threonine protein kinase ATM (ataxia telangiectasia mutated), but it may also be activated by DNA-dependent protein kinase (DNA-PKcs) when DNA damage occurs during mitosis. Fanconi anaemia (FA) group D2 or FANCD2 is a protein of the FA signalling transduction pathway. FANCD2 also function as a transducer of ATM signalling; the phosphorylation of FANCD2 at Ser222, initiated by ATM, contributes to arresting cells in the S phase of the cell cycle. Taking into account the current knowledge about the mechanisms of action of these two proteins as well as the mechanisms of action of CIT, we considered it interesting to investigate the potential involvement of these two proteins in CIT mode of action on different cells of human origin.

  1. Differences in the response to CIT treatment between cell lines is not a new observation (as reviewed in [17 and 20]). Although the present study may be the first to report CIT effects on the lung cancer cell line A459, it is not the first study with lung (i.e. V79) cells.

We agree with the reviewer, response to CIT treatment in cell lines is not a new observation; we cited the corresponding literature and expanded the discussion on genotoxic potential (please see point 3-4). However, to our knowledge, in our study genotoxicity was tested for the first time in A549 cells, as we stated in the manuscript. “Moreover, to the best of our knowledge, our findings are the first which showed genotoxic effect on lung cancer cells”.

  1. The „limitations of this type of research“ (line 262 and ff. text) is not really adequate (and mainly speculative). Instead, the Discussion should also address other published data on the genotoxic potential of CIT (see above remark #3).

Thank you for this observation. We have expanded the discussion to address some additional published data as suggested (please see explanations under point 3-4.) Yet, we believe that the paragraph about the limitations of the study gives a clear picture that we are aware of all the additional experiments that need to be done in future research on the investigated topic.

  1. What is missing also is a reflection how concentrations tested in vitrorelate to in vivo CIT levels reported in plasma or serum of mycotoxin-exposed humans. Such data can be found in biomonitoring studies as reviewed e.g. by Ali N et al. doi: 10.1007/s00204-019-02570-y) and the last Discussion section of an earlier in vitro study (Ref. [17]).

We thank Reviewer for the suggestion and hope that the following text will contribute to the quality of the manuscript:

The review on occurrence of CIT in various food commodities [36] showed a wide dissemination of CIT in different continents, such as Europe, Asia, America, and Africa. Cereal-based food and red rice food supplements seems to be the main source of exposure to CIT; the highest levels of CIT were found in maize in Serbia (mean 950 µg/kg), in Tom bran from Nigeria (up to 1173 µg/kg) and in red rice from China (up to 44240). Apples, analysed in Portugal were also highly contaminated with CIT (320-920 µg/kg). In Germany, France and Croatia CIT were detected in cereals with mean concentrations 10-30 µg/kg. However, CIT levels in food are insufficient for intake assessment. Ali and Degen [37] compiled data on human exposure to CIT in different countries by LC-MS/MS analysis of urine; in Belgium and Germany concentrations in urine were much lower (0.016 and 0.03 ng/mL) than in Bangladesh (0.59 ng/mL) and Nigeria (5.96 ng/mL); calculated provisional daily intake (PDI) of CIT in EU countries were lower than tolerable daily intake (TDI, 0.2 μg/kg b.w.) set by EFSA [8], while in the Bangladesh and Nigerian cohort PDI exceeded the TDI set for this mycotoxin. Based on the effective CIT concentrations applied in the in vitro models, it is difficult to predict intracellular accumulation in target organs resulting from exposure to CIT in humans. In vitro experiments in MDCK epithelial kidney cells indicated that CIT forms a stable complex with human serum albumin which may influence accumulation in target organs [27]. Yet, a half-life of CIT estimated from plasma concentration-time profiles in humans indicated a short half-life of CIT (~9 h) without a cumulative potential [38]. Taken together, humans are chronically exposed to CIT trough food consumption and occasionally its levels may exceed TDI indicating that humans might be exposed to genotoxic concentrations of CIT. Additionally to oral intake, humans may be exposed to this mycotoxin through respiratory route as CIT was detected in house dust [6]. Some earlier studies have shown that mycotoxin inhalation may be at least 10 times more toxic than oral intake due to quick absorption [39,40]. Toxicological relevance of CIT intake trough respiratory route remains to be determined in future studies.

Corrections

Abstract, line 10: delete ’culture’ before lines

Corrected.

Line 18: (edit) ... CIT at levels near the IC50 values induces ...

Corrected.

Line 66 and heading of Table 1: It can be argued whether „Metabolic activity“ is the best term or rather „Cytotoxicity“ ...

Changed to “cytotoxicity”, both in the text and on the Fig.1.

Fig. 2 Legend should state that data for TL and TI were log10 transformed.

Corrected.

Lines 109, 127, 129 and 131: replace „faze“ by the correct term „phase“ .......

Corrected.

Line 156: ...CIT-treatment, than (not: then) .....

Corrected.

Line 207: correct the spelling of „below“ (not: bellow)

Corrected.

Line 272 (edit): ... concentrations near and sometimes below IC50 values

Corrected.

Reviewer 3 Report

Comments and Suggestions for Authors

The manuscript describes the effects in vitro of two cell lines after exposure to different concentrations of citrinin. The experimental methods used are: MTT for cell viability, Comet assay for DNA damage, flow cytometry for cell cycle analysis, and an ELISA kit for Chk2 and FANCD2 phosphorylation and protein expression level. The innovative part is the ELISA experiment. The results are not surprising, especially after highly concentrated exposures.

My biggest concern about this research is the utility for the scientific community. How are related the exposure concentrations in vitro with real life? Which is the usual citrinin intake for humans? And for animals? Does it vary among the different regions in the world? There are several toxicokinetic studies that might help authors calculate seriously the citrinin intake that would be necessary to reach the toxic concentrations found in this research. I think this calculus and explanations are necessary in the manuscript for understanding the relevance of this research and justify its publication.

Links to revise for toxicokinetics:

https://doi.org/10.1016/j.fct.2020.111365

https://doi.org/10.1007/s00204-019-02570-y

https://doi.org/10.3390/toxins7124871

In Material and methods, the Reagents section is missing.

L300: Why do authors starve the cells before the exposure for the MTT proliferation assay? In my opinion, this step stresses the cells and it is not correlated with real life conditions. It is not usual at all in MTT protocols. Please, explain it carefully in the manuscript.

Minor comments:

Change the word treatment for exposure in the whole text, more suitable in toxicology area.

L6-7: Italics for Penicillium, Aspergillus and Monascus

L290: superíndex for the +2.

L300: use h instead of hours. It is usual to write the International units abbreviated.

L306: MTT tetrazolium salt reagent concentration is not explained.

L332: Na2EDTA, 2 in subindex

L421-423: As authors mention, according to the latest European regulation, citrinin is only limited in food supplements based on rice fermented with red yeast Monascus purpureus at 100 microg/kg. I think the legislation that should be cited is the latest: Commission Regulation (EU) 2023/915 of 25 April 2023 on maximum levels for certain contaminants in food and repealing Regulation (EC) No 1881/2006.

Figure 2 is pixelated, please increase its quality.

Figure 3. It seems weird not to have significance in A549 G2 and S phases at 100 microM. Have you double checked the statistics?

Author Response

Citrinin provoke DNA damage and cell cycle arrest related to Chk2 and FANCD2 checkpoint proteins in hepatocellular and adenocarcinoma cell line

We hugely thank the reviewers and the editor for giving us the opportunity to respond to the reviewers’ suggestions, correct the manuscript accordingly and give the improved version for the reconsideration for publication in Toxins. We are grateful to all the reviewers for taking the time to review this manuscript. We have taken into consideration all the kind suggestions to improve the quality of the paper. Please find the detailed responses below and the corresponding revisions/corrections highlighted in the re-submitted files.

Point-by-point response to Comments and Suggestions for Authors - Reviewer 3:

My biggest concern about this research is the utility for the scientific community. How are related the exposure concentrations in vitro with real life? Which is the usual citrinin intake for humans? And for animals? Does it vary among the different regions in the world? There are several toxicokinetic studies that might help authors calculate seriously the citrinin intake that would be necessary to reach the toxic concentrations found in this research. I think this calculus and explanations are necessary in the manuscript for understanding the relevance of this research and justify its publication.

Links to revise for toxicokinetics:

https://doi.org/10.1016/j.fct.2020.111365

https://doi.org/10.1007/s00204-019-02570-y

https://doi.org/10.3390/toxins7124871

We thank the reviewer for the suggestion. The aim of our study was to investigate mechanism of CIT action related to genotoxicity, altered Chk2 and FANCD2 phosphorylation and/or expression and induction of cell cycle arrest. To study that mechanism, we used two concentrations significantly lower than the IC50 and one higher concentration close to the IC50. The same approach was used in experiments aimed to study CIT mechanisms of action on the in vitro models cited in the manuscript (e.g. HepG2, V79, L02 cells), and similar concentrations were used as in the present study. We discussed our results related to genotoxic action in comparison with those studies. However, based on the effective CIT concentrations that provoked significant DNA damage in the in vitro models, it is difficult to predict intracellular accumulation in target organs resulting from exposure to CIT in humans. Nevertheless, respiratory intake of CIT was not studied yet. We gave some reflection regarding recent studies on the occurrence of CIT in food around the world as well as the finding of CIT in human urine. Once again, we thank Reviewer for the suggestion and hope that the following text will contribute to the quality of the manuscript:

The review on occurrence of CIT in various food commodities [36] showed a wide dissemination of CIT in different continents, such as Europe, Asia, America, and Africa. Cereal-based food and red rice food supplements seems to be the main source of exposure to CIT; the highest levels of CIT were found in maize in Serbia (mean 950 µg/kg), in Tom bran from Nigeria (up to 1173 µg/kg) and in red rice from China (up to 44240). Apples, analysed in Portugal were also highly contaminated with CIT (320-920 µg/kg). In Germany, France and Croatia CIT were detected in cereals with mean concentrations 10-30 µg/kg. However, CIT levels in food are insufficient for intake assessment. Ali and Degen [37] compiled data on human exposure to CIT in different countries by LC-MS/MS analysis of urine; in Belgium and Germany concentrations in urine were much lower (0.016 and 0.03 ng/mL) than in Bangladesh (0.59 ng/mL) and Nigeria (5.96 ng/mL); calculated provisional daily intake (PDI) of CIT in EU countries were lower than tolerable daily intake (TDI, 0.2 μg/kg b.w.) set by EFSA (2012), while in the Bangladesh and Nigerian cohort PDI exceeded the TDI set for this mycotoxin. Based on the effective CIT concentrations applied in the in vitro models, it is difficult to predict intracellular accumulation in target organs resulting from exposure to CIT in humans. In vitro experiments in MDCK epithelial kidney cells indicated that CIT forms a stable complex with human serum albumin which may influence accumulation in target organs [27]. Yet, a half-life of CIT estimated from plasma concentration-time profiles in humans indicated a short half-life of CIT (~9 h) without a cumulative potential [38]. Taken together, humans are chronically exposed to CIT trough food consumption and occasionally its levels may exceed TDI indicating that humans might be exposed to genotoxic concentrations of CIT. Additionally to oral intake, humans may be exposed to this mycotoxin through respiratory route as CIT was detected in house dust [6]. Some earlier studies have shown that mycotoxin inhalation may be at least 10 times more toxic than oral intake due to quick absorption [39,40]. Toxicological relevance of CIT intake trough respiratory route remains to be determined in future studies.

In Material and methods, the Reagents section is missing.

It is worth mentioning that already in the first version of the manuscript we listed all the reagents and corresponding manufacturers in the section Methods. However, according to the reviewer's comment, we changed the way of listing the reagents, added a section Reagents (shown below) and changed the numbering of the following sections.

5.1 Reagents

Components for cell culture maintenance, including RPMI 1640, FBS, trypsin-EDTA, phosphate-buffered saline (PBS; Ca2+ and Mg2+ free), penicillin, and streptomycin, were from Lonza (Basel, Switzerland). CIT was purchased from Cayman Chemicals, Ann Arbor, MI, USA), while NaOH and NaCl used for comet test were from Kemika (Zagreb, Croatia). Colorimetric cell-based ELISA kits CytoGlow™ Chk2 (phosphoThr-68) and CytoGlow™ FANCD2 (phosphoSer-222) were from AssayBioTech, Fremont, CA, USA). If not stated otherwise, all the other reagents were purchased from Sigma-Aldrich (Steinheim, Germany).

L300: Why do authors starve the cells before the exposure for the MTT proliferation assay? In my opinion, this step stresses the cells and it is not correlated with real life conditions. It is not usual at all in MTT protocols. Please, explain it carefully in the manuscript.

As suggested, we added an explanation to the section Discussion:

Before exopure to CIT, cells were serum-starved. Serum starvation prior to exposure to toxin/chemical/drug etc.  is a common procedure, which allows synchronization of cells to the same cell cycle phase. In this regard, theoretically, the impact that cell cycle will have on cells’ response to treatment is removed. Therefore, it is easier to detect toxin mode of action.

Minor comments:

Change the word treatment for exposure in the whole text, more suitable in toxicology area.

Corrected.

L6-7: Italics for Penicillium, Aspergillus and Monascus

Corrected.

L290: superíndex for the +2.

Corrected.

L300: use h instead of hours. It is usual to write the International units abbreviated.

Corrected.

L306: MTT tetrazolium salt reagent concentration is not explained.

Corrected.

L332: Na2EDTA, 2 in subindex

Corrected.

L421-423: As authors mention, according to the latest European regulation, citrinin is only limited in food supplements based on rice fermented with red yeast Monascus purpureus at 100 microg/kg. I think the legislation that should be cited is the latest: Commission Regulation (EU) 2023/915 of 25 April 2023 on maximum levels for certain contaminants in food and repealing Regulation (EC) No 1881/2006.

Corrected. The reference Commission Regulation (EU) 2023/915 is under the number 5.

Figure 2 is pixelated, please increase its quality.

Thank you for pointing this out as indeed the resolution of the Fig. 2 was quite poor. Reviewer 1 made the same observation. We inserted new version of the Figure with better resolution.

Figure 3. It seems weird not to have significance in A549 G2 and S phases at 100 microM. Have you double checked the statistics?

Reviewer noticed this nicely. The asterisks marking the statistically significant differences somehow are missing on the figure, though those differences were calculated to be statistically significant and accordingly, described as such in the first submitted version of the manuscript, i.e., it is stated that: “Regarding A549, the smallest applied concentration of CIT (50 µM) showed no effect on cell-cycle distribution, while accumulation of cells in G2/M faze and reduction of cells in S faze was observed in both higher concentrations of CIT (100 and 200 µM).”

Thank you for pointing this out as this was mistake on the Fig. 3. We added the missing asterisks.

(a)

(b)

Round 2

Reviewer 2 Report

Comments and Suggestions for Authors

The authors have adequately addressed the points of critique in the revised version. With the corrections made and additional text for improved clarification, the manuscript can be considered almost suitable for publication. Yet, the referee recommends to make further amendments and corrections:

1)    The Discussion now contains important information on CIT exposure (also in relation to provisional tolerable intakes as set by EFSA). Yet, the authors’ statement in Line 480ff “Based on the effective CIT concentrations applied in the in vitro models it is difficult to predict intracellular accumulation in target organs resulting from exposure to CIT in humans.“ This wording is not wrong; however, in vitro-in vivo extrapolation was not the point made before by the referee who did suggest to provide some orientation how the applied in vitro (high µM !) concentrations compare to internal circulating levels of CIT in exposed humans (in the nM range). This difference should be made more explicit, for instance by adding a short sentence in line 482 on plasma levels reported in humans or elsewhere in the text. As an aside, the binding of CIT to plasma proteins is far lower than that of OTA and unlikely to give rise to a significant accumulation in cells.

2)     Line 487: correct spelling of “through“ and  ... exceed (add:) the TDI value ... (Note: the referee has doubts that even a 10-fold exceedance of the TDI would result in internal CIT levels giving rise to genotoxicity; yet the authors are free to speculate).

3)    Line 490: The authors refer here to two papers on T-2, a mycotoxin with far higher cytotoxicity than CIT, and thus likely to exert local toxicity at the site of contact. Yet the reader should not be mislead and apply this ’concept’ to CIT or mycotoxins in general. Thus, please reword the sentence, e.g.: Earlier studies have shown that inhalation of T-2 mycotoxin may be .... more toxic than oral intake [39, 40] (delete the rest since this is is not necessarily related to quick absorption, but also less efficient metabolism at the first site of contact).

4)    Line 492: correct spelling of “through“

Comments on the Quality of English Language

fine with me

Author Response

We hugely thank the reviewers and the editor for giving us the opportunity to respond to the reviewers’ suggestions, correct the manuscript accordingly and give the improved version for the reconsideration for publication Toxins. We are grateful to all the reviewers for taking the time to review this manuscript. Please find the detailed responses below and the corresponding revisions/corrections highlighted in the re-submitted files.

Point-by-point response to Comments and Suggestions for Authors - Reviewer 2:

Specific comments

1) The Discussion now contains important information on CIT exposure (also in relation to provisional tolerable intakes as set by EFSA). Yet, the authors’ statement in Line 480ff “Based on the effective CIT concentrations applied in the in vitro models it is difficult to predict intracellular accumulation in target organs resulting from exposure to CIT in humans.“ This wording is not wrong; however, in vitro-in vivo extrapolation was not the point made before by the referee who did suggest to provide some orientation how the applied in vitro (high µM !) concentrations compare to internal circulating levels of CIT in exposed humans (in the nM range). This difference should be made more explicit, for instance by adding a short sentence in line 482 on plasma levels reported in humans or elsewhere in the text. As an aside, the binding of CIT to plasma proteins is far lower than that of OTA and unlikely to give rise to a significant accumulation in cells.

Response 1. Corrected as follows: Taken together, humans are chronically exposed to CIT through food consumption and occasionally its levels may exceed TDI indicating that humans might be exposed to higher levels of CIT. Still, considering available data on CIT in plasma and urine of adults it is unlikely that micromolar concentrations will be reached in target organs. Additionally to oral intake, humans may be exposed to this mycotoxin through respiratory route as CIT was detected in house dust [6]. Some earlier studies have shown that inhalation of T-2 mycotoxin may be at least 10 times more toxic than oral intake.[39,40]. The toxicological relevance of CIT intake through respiratory route remains to be determined in future studies.

2) Line 487: correct spelling of “through“ and... exceed (add:) the TDI value ... (Note: the referee has doubts that even a 10-fold exceedance of the TDI would result in internal CIT levels giving rise to genotoxicity; yet the authors are free to speculate).

Response 2. Corrected.

3) Line 490: The authors refer here to two papers on T-2, a mycotoxin with far higher cytotoxicity than CIT, and thus likely to exert local toxicity at the site of contact. Yet the reader should not be mislead and apply this ’concept’ to CIT or mycotoxins in general. Thus, please reword the sentence, e.g.: Earlier studies have shown that inhalation of T-2 mycotoxin may be .... more toxic than oral intake [39, 40] (delete the rest since this is is not necessarily related to quick absorption, but also less efficient metabolism at the first site of contact).

Response 3. Corrected at point 1.

4) Line 492: correct spelling of “through“

Response 4. Corrected.

Reviewer 3 Report

Comments and Suggestions for Authors

The authors have modified the manuscript according to the suggestions sent.

Author Response

Point-by-point response to Comments and Suggestions for Authors - Reviewer 3:

The authors have modified the manuscript according to the suggestions sent.

We thank Reviewer 3 for the comment.